# Epidemiological Characteristics of Carbapenem-Resistant *Enterobacterales* in Japan: A Nationwide Analysis of Data from a Clinical Laboratory Center (2016–2022)

**DOI:** 10.3390/pathogens12101246

**Published:** 2023-10-16

**Authors:** Kentarou Takei, Miho Ogawa, Ryuji Sakata, Hajime Kanamori

**Affiliations:** 1Department of Infectious Diseases, Internal Medicine, Tohoku University Graduate School of Medicine, Sendai 980-8575, Japan; kanamori@med.tohoku.ac.jp; 2Department of Bacteriology, BML Inc., Kawagoe 350-1101, Japan

**Keywords:** carbapenem-resistant *Enterobacterales*, infection surveillance, multidrug-resistant organisms, carbapenemase producing

## Abstract

In Japan, nationwide epidemiological surveys on carbapenem-resistant *Enterobacterales* (CREs), including comprehensive information, are scarce, with most data available only through public reports. This study analyzed data on the *Enterobacterales* family collected from nationwide testing centers between January 2016 and December 2022, focusing on isolates that met the criteria for CRE in Japan based on drug susceptibility. We investigated 5,323,875 *Enterobacterales* isolates of 12 different species; among 4696 (0.09%) CRE strains, the proportion of major CRE isolates was as follows: *Escherichia coli*, 31.3%; *Klebsiella pneumoniae*, 28.0%; *Enterobacter cloacae*, 18.5%; and *Klebsiella aerogenes*, 6.7%. Moreover, over a 7-year period, *Providencia rettgeri*, *E. cloacae*, *K. aerogenes*, and *K. pneumoniae* demonstrated relatively high CRE percentages of 0.6% (156/26,185), 0.47% (869/184,221), 0.28% (313/110,371), and 0.17% (1314/780,958), respectively. The number of CRE strains isolated from different samples was as follows: urine, 2390; respiratory specimens, 1254; stool, 425; blood, 252; others, 375. In the broader context, including colonization, the predominant isolates of CREs collected at nationwide testing centers are *E. coli* and *K. pneumoniae*. Furthermore, recently, attention has been directed toward less common CRE species, such as *Klebsiella oxytoca* and *Providencia rettgeri*, and thus, it might be necessary to continue monitoring these less common species.

## 1. Introduction

Carbapenem-resistant *Enterobacterales* (CREs) are multidrug-resistant organisms that cause healthcare-associated infections, posing a public health threat [1]. Outbreaks of CRE infections frequently occur in healthcare facilities, leading to an increased number of deaths, events of illnesses, and medical costs resulting from this type of antimicrobial resistance (AMR) [2].

Carbapenem resistance in *Enterobacterales* is primarily due to the production of carbapenemases (CPs), which break down carbapenem agents classified under classes A, B, and D according to the Ambler classification, or the species may be non-CP-producers [3]. The representative of Class A CPs is *Klebsiella pneumoniae* carbapenemase (KPC), while that of Class B is New Delhi metallo-β-lactamase (NDM) [4,5]. The possession of these CPs is a topic of great importance due to high mortality rates and horizontal transfer resulting from CREs [6,7]. Moreover, based on review reports from various regions around the world, *K. pneumoniae* is commonly reported as the predominant causative agent of both CRE colonization and infection, although in some cases, *Enterobacter* spp. may be predominant [1,8].

The World Health Organization (WHO) has developed an action plan to handle AMR in microbes and has described the accumulation of knowledge and evidence through research as one of the key pillars of AMR control [9]. The monitoring and investigation of the drug resistance of CREs are carried out worldwide, as well as investigations into the presence of resistance genes in regional CREs [1,10,11,12]. In Japan, two nationwide surveillance systems for antibiotic-resistant bacteria exist under the purview of the Ministry of Health, Labour, and Welfare (MHLW). The first is the National Epidemiological Surveillance of Infectious Diseases (NESID), which focuses on CRE infection cases identified based on the Infectious Disease Control Law. The second is Japan Nosocomial Infections Surveillance (JANIS), which focuses on hospitals actively participating in infection control programs linked with healthcare reimbursement and accounts for approximately one-quarter of medical institutions in Japan [13,14]. These public surveillance systems reported *Klebsiella aerogenes* and *Enterobacter cloacae* as the predominant CRE species in Japan [15], whereas previous research conducted in specific regions of Japan has reported that CREs identified in colonization cases primarily consisted of *Escherichia coli* and *K. pneumoniae* [16], suggesting a difference in CRE species between infection and colonization cases. The overall landscape of isolated and cultured CREs in Japan, including colonization, has not been elucidated.

Various healthcare institutions, particularly those lacking their own testing facilities, outsource bacterial tests. Testing companies receive and handle specimens, including those related to infection or colonization. However, thus far, a comprehensive overview of the overall spectrum of bacterial species isolated and cultured by these testing companies has not been reported. Moreover, it has not been determined which bacterial species exhibit a high prevalence of CREs among those tested at testing centers.

In this study, by analyzing nationwide data, including colonization cases from a domestic testing center, we conducted an epidemiological investigation of *Enterobacterales* species that met the criteria for CRE in Japan. Such a nationwide study, which includes data collected from a private center, has not been previously reported.

## 2. Materials and Methods

### 2.1. Bacterial Database

Bacterial data were obtained from a major domestic laboratory (BML Inc., Tokyo, Japan). BML offers bacterial testing services in all prefectures across Japan. All bacterial data were categorized based on bacterial species, material type, and drug susceptibility. Samples from which the data were obtained from various places were submitted for bacterial cultivation and identification between January 2016 and December 2022. *Enterobacterales* in the samples were identified and subjected to antimicrobial susceptibility testing using the Microscan WalkAway^®^ system (Beckman Coulter, Brea, CA, USA) with accompanying panels of the Microscan Neg^®^ series (Neg Combo EN 4J and Neg MIC EN 2J) (Beckman Coulter, Brea, CA, USA). The bacterial solution used for this test was prepared using the prompt inoculation method [17], according to the guidelines of the Clinical and Laboratory Standards Institute (CLSI) [18]. CRE was defined based on the Infectious Disease Control Law in Japan—*Enterobacterales* isolates with a minimum inhibitory concentration (MIC) of ≥2 µg/mL for meropenem, ≥2 µg/mL for imipenem, and ≥64 µg/mL for cefmetazole.

### 2.2. Analysis of CRE Data

All *Enterobacterales* species were listed, and species with more than 20 CRE isolates were selected for analysis. Among *Enterobacterales* species, pathogenic *E. coli*, *Yersinia*, *Salmonella*, and *Shigella* species were excluded from the study because these species were most likely submitted for testing by employees who primarily handle food products. Individual and hospital identifications were not performed, and duplicate samples were collected.

The CRE percentage was calculated as follows: (CRE isolates count/total bacterial isolates count) × 100 percent (%). The specimens associated with the detected CRE isolates were also investigated. The samples were categorized into three groups: urological samples, mainly including urine; respiratory tract samples, including sputum, nasal discharge, and throat swabs; and stool samples. All other samples were grouped under “Others”, which encompassed pus, ear discharge, vaginal secretions, wound site fluids from bedsores, drain fluids, and others.

### 2.3. Statistical Analysis

The CRE percentage was analyzed using an analysis of variance (ANOVA), and the Tukey–Kramer HSD test was used for the pairwise comparisons of mean values. The analyses were performed using JMP Pro 16 (SAS Institute, Cary, NC, USA). Results with *p* < 0.05 were considered statistically significant.

## 3. Results

### 3.1. Overview of Enterobacterales Species in This Study

In total, 5,323,875 *Enterobacterales* strains were isolated between 2016 and 2022. The analysis included 12 bacterial species, listed in descending order based on the number of isolates: *E. coli* (57.2%) > *K. pneumoniae* (14.7%) > *Proteus mirabilis* (6.1%) > *Serratia marcescens* (4.0%) > *Klebsiella oxytoca* (3.9%) > *Enterobacter cloacae* (3.5%) > *Morganella morganii* (2.6%) > *Citrobacter koseri* (2.3%) > *Citrobacter freundii* (2.1%) > *Klebsiella aerogenes* (2.1%) > *Providencia stuartii* (1.3%) > *Providencia rettgeri* (0.5%). *E. coli* and *K. pneumoniae* accounted for more than 50% of the total bacterial population. Throughout the 7-year period, the proportion of each bacterial species remained relatively constant.

### 3.2. Epidemiological Trends of CRE Species in Japan

Out of 5,323,875 *Enterobacterales* isolates, there were a total of 4696 CRE isolates. The number of CRE isolates and CRE percentages per year in 12 bacterial species from 2016 to 2022, respectively, were as follows: 416 isolates (0.06%), 937 isolates (0.13%), 833 isolates (0.11%), 669 isolates (0.09%), 641 isolates (0.08%), 704 isolates (0.09%), and 496 isolates (0.06%). The proportion of each bacterial species for the 7-year period was as follows: *E. coli*, 1469 (31.3%); *K. pneumoniae*, 1314 (28.0%); *E. cloacae*, 869 (18.5%); *K. aerogenes*, 313 (6.7%); *P. rettgeri*, 156 (3.3%); *S. marcescens*, 134 (2.9%); *C. koseri*, 118 (2.5%); *K. oxytoca*, 114 (2.4%); *M. morganii*, 66 (1.4%); *P. stuartii*, 66 (1.4%); *C. freundii*, 49 (1.0%); and *P. mirabilis*, 28 (0.6%). Therefore, *E. coli* was the most prevalent species (Figure 1).

For the four representative CRE species with the highest number of isolates, the number of CRE isolates and CRE percentage are presented in Figure 2. In the regression analysis of CRE isolates, *E. coli* and *K. pneumoniae* showed a decreasing trend, while *E. cloacae* and *K. aerogenes* also exhibited an increasing trend. However, the *p*-values for statistical significance in the regression analysis were 0.372, 0.638, 0.221, and 0.194, respectively, indicating no statistical significance.

The CRE percentage for the 7 years was as follows: *P. rettgeri*, 0.6%; *E. cloacae,* 0.47%; *K. aerogenes*, 0.28%; *K. pneumoniae*, 0.17%; *C. koseri,* 0.1%; *P. stuartii*, 0.1%; *S. marcescens*, 0.06%; *K. oxytoca*, 0.06%; *E. coli*, 0.05%; *M. morganii*, 0.05%; *C. freundii,* 0.04%; and *P. mirabilis*, 0.01%. Notably, the number of CRE isolates and CRE percentage of *P. rettgeri* detected over the three years from 2017 to 2019 were, respectively, 36 isolates (1.02%), 36 isolates (1.09%), and 37 isolates (1.05%). No other species besides *P. rettgeri* had a CRE percentage exceeding 1%.

In a comparison of the CRE percentage using the Tukey–Kramer HSD test, *P. rettgeri* showed a significantly higher CRE percentage than *P. mirabilis*, *C. freundii*, *E. coli*, *M. morganii*, *K. oxytoca*, *S. marcescens*, *C. koseri*, *P. stuartii*, *K. pneumoniae* (*p* < 0.0001), and *K. aerogenes* (*p* = 0.0004). Moreover, *E. cloacae* exhibited a significantly higher CRE percentage than *C. freundii*, *E. coli*, *M. morganii*, *K. oxytoca*, *S. marcescens*, *P. stuartii*, *C. koseri* (*p* < 0.0001), and *K. pneumoniae* (*p* = 0.0037). The CRE percentage of *K. aerogenes* was significantly higher than that of *P. mirabilis* (*p* = 0.0125), but no statistically significant differences were observed between other bacterial species.

### 3.3. Relationship between the Submitted Samples and CRE Species

Among the total submitted samples with CRE isolates, the proportion of urine and urogenital samples (2390 specimens) was the highest, followed by respiratory and throat specimens (1254 specimens), stool (424 specimens), blood (252 specimens), and others (375 specimens). Of the 12 bacterial species, 10 were primarily isolated from urine and urogenital samples. The ranking of stool, blood, and other samples varied according to bacterial species. *P. mirabilis* and *E. cloacae* were mainly isolated from the respiratory tract and throat. Details of the sample proportions of representative CREs are provided in Figure 3. For each CRE species, the proportion of blood samples suggesting evident infections was as follows: *K. oxytoca*; 12.3% (14/114); *S. marcescens* 9.7% (13/134); *K. aerogenes* 8.3% (26/313); *C. freundii* 8.2% (4/49); *E. cloacae*, 7.7% (67/869); *K. pneumoniae*, 4.7% (62/1313); *M. morganii*, 4.5% (3/66); *E. coli*, 4.0% (59/1469); *P. mirabilis*, 3.6% (1/28); *C. koseri*, 1.7% (2/118); *P. rettgeri*, 0.6% (1/156); and *P. stuartii*, 0% (0/66). Pairwise comparisons using the Tukey–Kramer HSD test indicated that the proportion of *K. oxytoca* was significantly higher than that of *P. stuartii* (*p* < 0.05), but no significant differences were observed in the proportions of other species.

## 4. Discussion

In this study, we examined the prevalence and distribution of 12 CRE species, which were identified and tested for antimicrobial susceptibility from 2016 to 2022 at a nationwide test center. We aimed to focus on the detection of CRE isolates in specimens submitted for testing, rather than targeting infectious agents.

The NESID and JANIS primarily monitor CRE infections and have consistently reported a high prevalence of *K. aerogenes* in cases of CRE infection [15,19]. However, in the present study, *K. aerogenes* was the least prevalent among the four major bacterial isolates, exhibiting a lower CRE isolate count and CRE percentage. This study reflects the results of isolating and culturing clinical specimens, including a substantial number of colonization cases; therefore, this different approach may have contributed to the differences between the public reports and the results of this study. Moreover, variations are observed even in the predominant specimen types; while the JANIS reports primarily involve respiratory specimens, our study predominantly involved urine specimens, with a lower frequency of blood specimens indicating overt infections [14].

In the following sections, we discuss each bacterial species that was included in the study.

### 4.1. Escherichia coli and Klebsiella pneumoniae

In recent years, reports have consistently identified *K. pneumoniae* as the predominant CRE isolate in major countries worldwide [1,12,20]. While there has been growing concern over the increase in carbapenem-resistant *E. coli* in Europe, it is important to note that *E. coli* is not currently considered the primary CRE isolate [21]. Even within Japan, the frequency of *E. coli* as a causative agent of CRE infections is not particularly high. According to domestic reports, infections caused by carbapenem-resistant *E. coli* account for only 6.5–10% of all CRE infections, making up a relatively low proportion [15]. However, when considering colonization, different possibilities have been suggested. In a previous report, the CRE percentage in long-term care hospitals was significantly higher (14.9%) than that in acute care hospitals (3.6%), and *E. coli* was the most predominant CRE isolate [16]. In our study as well, *E. coli* comprised 1469 carbapenem-resistant isolates and was the most common CRE bacterial species. There are fewer reported cases of CRE infections; however, when considering both colonization and isolations in laboratory settings, *E. coli* is possibly the most prevalent in terms of the total number of occurrences. In addition, although our study did not specifically investigate CP-Enterobacterales, CPs are common among *E. coli* that exhibit carbapenem resistance [16,22]. As *E. coli* is rarely the causative agent of CRE infections, the carbapenem-resistant *E. coli* isolates identified in this study are suggested to represent colonization to some extent. One of the reasons for *E. coli* being the most prevalent species in this study is speculated to be the distribution of facilities targeted by testing companies. Namely, it could reflect the test results from small- and medium-sized hospitals, long-term care facilities, and clinics without their own testing facilities. JANIS investigates not only infection cases but also colonization, unlike NESID. However, a notable absence of participation of small-scale hospitals with fewer than 200 beds, constituting over 50% of the inpatient beds, is observed [14].

The above conclusions for *E. coli* can also be applied to *K. pneumonia*; *K. pneumoniae* is the second most frequently observed isolate of CRE in localized regional surveillance for CRE colonization within Japan, following *E. coli* [16]. These findings align with the results of the present study. In our study, the CRE percentage for *K. pneumoniae* was 0.17%, more than three times higher than that for *E. coli*, and the number of CRE isolates was the second highest and close to that of *E. coli*, although a decreasing trend was observed for *K. pneumoniae*, similar to that seen for *E. coli*. Similar to *E. coli*, carbapenem-resistant *K. pneumoniae* is likely to be a CP species [16]. *K. pneumoniae* is one of the most studied bacteria for carbapenemase production, and KPC and NDM are the most prevalent CPs in neighboring countries [23,24]. In Japan, the predominant type of carbapenemase is the IMP type [15,25].

### 4.2. Enterobacter cloacae and Klebsiella aerogenes

*Klebsiella aerogenes* was originally classified under the *Enterobacter* genus and was reclassified under the *Klebsiella* genus in 2019 owing to its genetic similarities [26]. However, infections caused by *K. aerogenes* and *E. cloacae* exhibit similarities in patient profiles, occurrence, and prognoses [27]. According to nationwide surveys in Japan, the frequency of infections caused by carbapenem-resistant *K. aerogenes* and *E. cloacae* is the highest, unlike the global trend [15]. This finding is consistent with those of other domestic studies [28,29]. The CRE percentages obtained in this study were 0.28% (313/110,371) for *K. aerogenes* and 0.47% (869/184,221) for *E. cloacae*. Moreover, *E. cloacae* had a significantly higher CRE percentage than eight other bacterial species, namely, *C. freundii*, *E. coli*, *M. morganii*, *K. oxytoca*, *S. marcescens*, *P. stuartii*, *C. koseri,* and *K. pneumoniae*. No statistically significant difference was observed, but both *E. cloacae* and *K. aerogenes* show an increasing trend, warranting attention to future developments (Figure 2).

Regarding the mechanism of carbapenem resistance, CP *K. aerogenes* isolates are rare in Japan [30]. In particular, *K. aerogenes* is mostly non-CP, and its resistance to carbapenems is greatly influenced by the use of antibiotics [31]. Consequently, it is more likely to be detected in relatively large-scale acute-care hospitals [28,29]. Non-CP *K. aerogenes* isolates were characterized by resistance to imipenem and susceptibility to meropenem in a previous study [29]. Resistance mechanisms involve the overproduction of AmpC β-lactamase, the action of efflux pumps, and the downregulation of porins [32,33,34,35]. Similarly, *E. cloacae* produces AmpC β-lactamase, but unlike *K. aerogenes*, *E. cloacae* is considered a representative CP species in Japan [28,29].

### 4.3. Minor Carbapenem-Resistant Enterobacterales

Monitoring programs of CRE species have primarily focused on common carbapenem-resistant bacteria, such as *E. coli*, *Enterobacter* spp., and *Klebsiella* spp. However, by focusing solely on these bacterial species, there is a risk of overlooking infections caused by relatively rare CRE species [36]. In this study, among the 12 bacterial species, the less common carbapenem-resistant bacteria were *S. marcescens*, *K. oxytoca*, *P. stuartii*, *P. rettgeri*, *C. koseri*, *C. freundii*, *M. morganii*, and *P. mirabilis*. These less common species have attracted attention in recent years in the context of CRE surveillance, although consolidated data on such rare species are scarce domestically. *Providencia* spp., *Proteus* spp., and *M. morganii* are intestinal bacteria that exhibit intrinsic resistance to imipenem, and their resistance to other carbapenems needs to be confirmed [37]. In Japan, considering intrinsic resistance to imipenem among previous species such as *M. morganii*, imipenem and cefmetazole resistance was set as a criterion for CRE, which may not necessarily align with definitions used in overseas settings. Moreover, because carbapenemase-coding genes can be transferred to other bacterial species through horizontal transfer via plasmids, CP is more important in less common carbapenem-resistant species [38,39,40].

The proportion of *Serratia. marcescens* as a causative agent of CRE infections in Japan ranges from 3.4% to 5% [15]. The CRE percentage for *S. marcescens* in the present study was 0.06% over 7 years. There is no indication of an increase in the number of isolates or CRE percentage for carbapenem-resistant *S. marcescens* in this study. A previous, relatively large-scale study that included 28 university hospitals in Japan reported that carbapenem-resistant *S. marcescens* isolates were non-CP, suggesting that many carbapenem-resistance mechanisms in *S. marcescens* are non-carbapenemase-related [28].

In this study, the total number of *Klebsiella oxytoca* isolates was nearly equivalent to that of *S. marcescens*. The number of CRE isolates was low, with 114 isolates showing a low CRE percentage. There have been no nationwide epidemiological survey reports of carbapenem-resistant *K. oxytoca* in Japan. The first report on CP *K. oxytoca* was published in 2015, and it has recently gained attention in the country [41]. A notable characteristic of *K. oxytoca* isolates obtained in our study is the high proportion of blood culture specimens (12.3%). This finding suggests that they cause apparent infections with potentially poor prognoses. However, detailed information is lacking owing to the limited number of domestic reports. Similar to *S. marcescens*, there is no indication of an increase in the number of isolates or CRE percentage for *K. oxytoca*.

*Providencia stuartii* and *P. rettgeri* are the primary causative agents of urinary tract infections and food poisoning [42,43]. These have previously been reported in patients with long-term indwelling urinary catheters [44]. In the present study, *P. stuartii* and *P. rettgeri* were the only *Providencia* species detected. Although the total number of *P. rettgeri* isolates was 26,185, which was the lowest among the 12 species, the CRE percentage was the highest at 0.6%. A notable feature of *P. rettgeri* is its high CRE percentage, which was significantly higher than that of 10 other bacterial species. The CRE percentage of *P. rettgeri* exceeded 1% from 2017 to 2019. Due to the low number of isolates of *P. rettgeri*, it cannot be ruled out that specific facilities or regional outbreaks are being observed because the relative number of CRE isolates is low. Although there is no evidence to suggest that monitoring programs have focused on carbapenem-resistant *P. rettgeri* in Japan during the study period, there have been reports of IMP-type metallo-β-lactamase-producing *P. rettgeri* in Japan during the study period, indicating a possible correlation [45].

In the present study, there were 120,754 and 113,090 *Citrobacter. koseri* and *C. freundii* isolates. Although the number of isolates was nearly the same, the number of carbapenem-resistant *C. koseri* isolates was high (118 isolates) compared to 49 carbapenem-resistant *C. freundii* isolates, and the CRE percentage was also higher for *C. koseri*. In previous domestic reports on infectious diseases, *C. freundii* has been more frequently observed, suggesting that *C. koseri* has a higher colonization percentage than infection percentage [15,28]. *C. koseri* and *C. freundii* typically have low isolate numbers and CRE percentages each year, suggesting that they are well controlled.

*Morganella morganii* is a chromosomal AmpC β-lactamase-producing bacterium that exhibits intrinsic resistance to imipenem [37]. However, the number of *M. morganii* isolates meeting the criteria for CRE in Japan is limited. According to the NESID, there were no reports of carbapenem-resistant *M. morganii* from 2017 to 2020. In a relatively large-scale study involving 28 university hospitals, 10 cases of non-CP carbapenem-resistant *M. morganii* infection were recorded [28]. Carbapenem-resistant *M. morganii* also has a low number of isolates and a low CRE rate, suggesting that it is well controlled.

*Proteus mirabilis* can cause various human infections, including wound, eye, digestive system, and urinary tract infections. It also causes catheter-associated urinary tract infections (CAUTIs) in the catheterized urinary tract [46]. Although there is a possibility that *P. mirabilis* produces carbapenemases, it is more likely that its carbapenem resistance is due to non-CP mechanisms [47]. In this study, carbapenem-resistant *P. mirabilis* accounted for the lowest number of isolates (28 isolates) over a span of 7 years, and the CRE percentage was less than 0.01%, indicating that resistance to carbapenems is not a major concern.

This study has several limitations. First, the distribution of bacterial isolates might be biased even within the same country due to variations in population density, average age, and other factors. Nevertheless, this is the first epidemiologic study utilizing a nationwide database including CRE infection and colonization in the extensive scope of testing in Japan. Second, individual identification of bacterial isolates was not conducted, suggesting that our database may include multiple redundant isolates from the same individual. However, carbapenem-resistant *E. coli* and *K. pneumoniae*, due to their much higher numbers, are unlikely to be affected, unlike the minor CRE species, by duplication. Third, as we screened for CRE based on Japanese criteria, they do not correspond to CREs defined in other countries’ guidelines (e.g., Clinical and Laboratory Standards Institute (CLSI) or European Committee on Anti-microbial Susceptibility Testing (EUCAST) [9,48]). Fourth, because we did not conduct genetic tests or genome analyses, the mechanisms underlying antibiotic resistance (e.g., carbapenemases, extended-spectrum β-lactamases, and AmpC β-lactamases) were not investigated.

## 5. Conclusions

We investigated the number and CRE percentage of *Enterobacterales* species in samples, including colonization, collected at the nationwide testing center from 2016 to 2022. We found that *E. coli* was the predominant CRE species, followed by *K. pneumoniae*. These two species together accounted for over 50% of the total CRE isolates in our study. Small and medium-sized hospitals, long-term care facilities, and clinics without their own testing facilities may rely on testing companies. In specimens requested from these healthcare facilities, carbapenem-resistant *E. coli* and *K. pneumoniae* could be frequently encountered. In addition, carbapenem resistance in *E. coli* and *K. pneumoniae* is likely to be due to CP. Considering the frequent movement of patients between healthcare facilities, CP-CREs raise concerns about horizontal gene transfer to other bacterial species. More attention should be paid to the potential spread of CREs in these healthcare institutions, and monitoring colonization will also be necessary to prevent the proliferation of CRE species.

## Figures and Tables

**Figure 1 pathogens-12-01246-f001:**
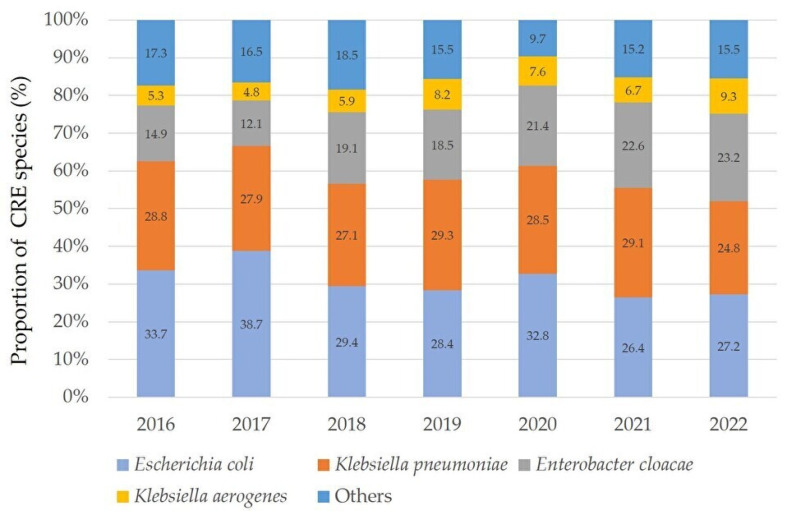
Annual frequency of representative CRE species from 2016 to 2022. In 2019 and 2021, *K. pneumoniae* was the most prevalent, but in other years, *E. coli* was the most common. The combined total of carbapenem-resistant *E. coli* and *K. pneumoniae* accounted for over 50% annually. “Others” includes the following bacterial species: *S. marcescens*, *K. oxytoca*, *P. stuartii*, *P. rettgeri*, *C. koseri*, *C. freundii*, *M. morganii*, and *P. mirabilis*.

**Figure 2 pathogens-12-01246-f002:**
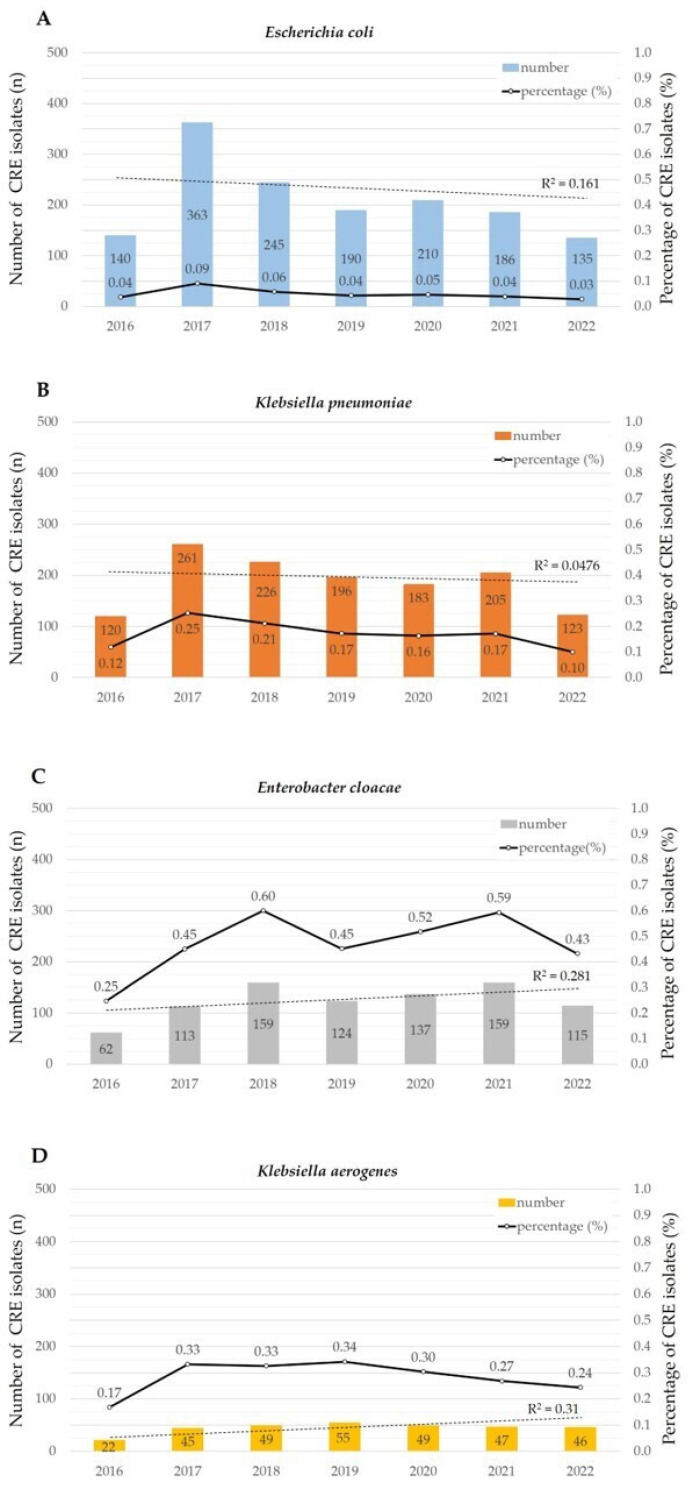
Temporal trends of four representative CRE species. The bar graph represents the number of CRE isolates, while the line graph illustrates the percentage of CRE isolates, The dashed line represents the regression analysis of isolated CRE counts (**A**–**D**). In 2017, both *E. coli* and *K. pneumoniae* shown a decreasing trend, as indicated in the regression analysis. *E. cloacae* and *K. aerogenes* showed significance was obtained in ANOVA. *E. cloacae* exhibited a significantly higher CRE percentage than *C. freundii*, *E. coli*, *M. morganii*, *K. oxytoca*, *S. marcescens*, *P. stuartii*, *C. koseri*, and *K. pneumoniae* for seven years. The CRE percentage of *K. aerogenes* was significantly higher than that of *P. mirabilis*.

**Figure 3 pathogens-12-01246-f003:**
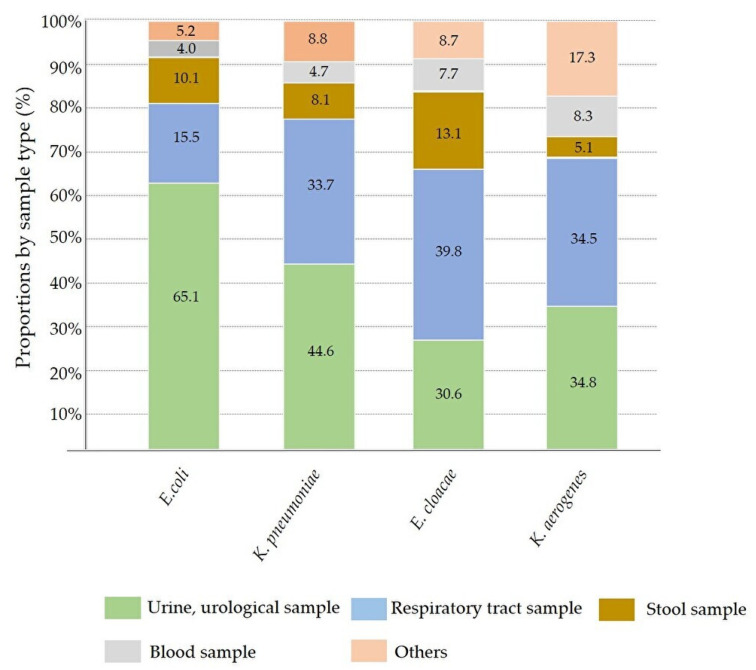
The proportion of submitted samples containing representative CRE isolates. *E. coli*, the most common CRE isolate, was found in approximately 65% of urinary samples. Unlike *E. coli*, *K. pneumoniae*, *E. cloacae*, and *K. aerogenes* showed a higher frequencies in upper respiratory and sputum samples.

## Data Availability

The datasets created and analyzed during the current study are not publicly available due to contain patient information and confidentiality obligations of the company. The dataset is owned by BML Inc., Tokyo, Japan.

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
