# Peer review of "Epidemiological Characteristics of Carbapenem-Resistant Enterobacterales in Japan: A Nationwide Analysis of Data from a Clinical Laboratory Center (2016–2022)"

_pathogens, 2023, doi:10.3390/pathogens12101246_

Round 1
Reviewer 1 Report
The study analzyed epidemiological characteristics of CRE in Japan. The study found that dominant CRE were E. coli and K. pneumoniae.
The aim of the study is novel, clear and well defined. The methods are su accurate and reproducible. The results are well represented and sound. The discussion and conclusions are balanced and adequately suppported by the data presented in the result section. The references are up to date and the quality of English language is excellent. The title and the abstract convey with what was found. The manuscript adhere with the relevant standards for reporting of the data.
MAJOR COMMENTS
1. In the majority of countries the microbiological laboratories perform PCR or immunochromatographic tests to determine the type of carbapenemase in the frames of routine laboratory diagnostic. This is important for the therapy choice because ceftazidime/avibactam usually shows excellent activity on OXA-48 whereas combinations with inhibitors are not effective on metallo-beta-lactamase producing organisms. I would suggest to include the types of carbapenemases in the analysis.
2. It is unusual that E. coli is the dominant species harbouring carbapenemase. In the majority of reports it is K. pneumoniae or E. cloacae. This should be explained in the discussion section.
3. It would be interesting to analyze if they coharboured ESBL as well. Enterobacterales are prone to carry multiple resistance genes on the same plasmid. This is particulary important for OXA-48 positive strains since this carbapenemase does not hydrolyze cephalosporins.
4. To my opinion there is too much listing of the data with little point.
5. I would suggest to divide the specimens into clinically relevant such as blood cultures, cerebrospinal fluid etc and colonization (rectum swab, nasopharyngela swab etc)
MINOR COMMENTS
6. Line 20: as follows
T
Author Response
-by-point responses to the Reviewer’s comments
Thank you for taking the time to review our manuscript and provide insightful and helpful comments, despite your busy schedule. We sincerely appreciate it. We have made revisions to the content of the paper based on your valuable comments. We kindly request you to review it once again. Thank you in advance for your assistance.
Reviewer1
Reviewer's suggestion or questions
- In the majority of countries the microbiological laboratories perform PCR or immunochromatographic tests to determine the type of carbapenemase in the frames of routine laboratory diagnostic. This is important for the therapy choice because ceftazidime/avibactam usually shows excellent activity on OXA-48 whereas combinations with inhibitors are not effective on metallo-beta-lactamase producing organisms. I would suggest to include the types of carbapenemases in the analysis.
Response to the reviewer
This study compiles the results of isolations, cultures, and antimicrobial susceptibility testing conducted by a testing company. The primary focus of this study is to understand the trends and overall picture of carbapenem-resistant bacterial species. Unfortunately, confirming the specific types of carbapenemases is beyond the scope of this study. We appreciate your understanding.
Reviewer's suggestion or questions
- It is unusual that E. coli is the dominant species harbouring carbapenemase. In the majority of reports it is K. pneumoniae or E. cloacae. This should be explained in the discussion section.
Response to the reviewer
We appreciate your careful observation. As you have mentioned, in a previous study conducted at our facility (1160 beds), the main species carrying carbapenemases were indeed E. cloacae and K. pneumoniae (reference 32). However, it is important to note that the distribution of carbapenem-resistant bacterial species can possibly vary by the size and type of the healthcare facilities. We have added a note in the discussion section to address the fact that E. coli is a major species, as per your suggestion.
Reviewer's suggestion or questions
- It would be interesting to analyze if they coharboured ESBL as well. Enterobacterales are prone to carry multiple resistance genes on the same plasmid. This is particulary important for OXA-48 positive strains since this carbapenemase does not hydrolyze cephalosporins.
Response to the reviewer
The suggestion to investigate the presence of ESBL is indeed intriguing. While the results of antimicrobial susceptibility testing suggest the presence of ESBL, it is important to note that genetic testing was not conducted in our study. Therefore, the coexistence of ESBL is not confirmed, and our conclusions are solely based on the results of antimicrobial susceptibility testing. This limitation has also been stated in the Limitation section of the study.
Reviewer's suggestion or questions
- To my opinion there is too much listing of the data with little point.
Response to the reviewer
I have removed the table, which was a listing of numerical data, and inserted a figure representing the annual percentage of predominant bacterial species, which is important information. I have also listed important numerical values in the Results section.
Reviewer's suggestion or questions
- I would suggest to divide the specimens into clinically relevant such as blood cultures, cerebrospinal fluid etc and colonization (rectum swab, nasopharyngela swab etc)
Response to the reviewer
Thank you for your important suggestion. We have accordingly included blood culture samples that provide clear indications of infectious diseases, but we did not have any examples of cerebrospinal fluid samples. As the purpose of the testing was not clear and it was challenging to distinguish between colonization and clinically relevant specimens, other than blood cultures, the categorization was limited to broad species groups.
Reviewer's suggestion or questions
MINOR COMMENTS
- Line 20: as follows
Response to the reviewer
We have accordingly made the necessary corrections.

Reviewer 2 Report
Review report of Journal Pathogens (MDPI)
The manuscript titled as “Epidemiological Characteristics of Carbapenem-Resistant Enterobacteriaceae in Japan: A Nationwide Analysis of Data from a Clinical Laboratory Center (2016–2022)” by Takei et al. conducted epidemiological investigation to Carbapenem-Resistant Enterobacteriaceae (CRE) in Japan utilizing data of 2016-2020 obtained from major domestic laboratory.
· The work is epidemiological in nature and the authors have basically emphasized on prevalence of different CRE species isolated from different types of submitted sample such as urine blood etc. only temporal distribution and proportion of different isolates are taken into account. An epidemiological survey include different aspects which must be taken into account such as demographics and geographic distribution. Wherein no such data is presented. Therefore for improving the article it is necessary to add such data.
· In abstract, line 10-11 needs rephrasing for better understanding.
· Introduction provides a solid foundation for the research. I would like to suggest inclusion of additional content in the introduction to provide a more comprehensive context for the study.
· In introduction section, add transition sentences between paragraphs to improve the flow and connection of ideas.
· In Introduction, highlight the research gap that your study intends to address. What specific aspects of CRE epidemiology in smaller healthcare settings have not been explored in previous research?
· In material method section, the author has not mentioned how the species are identified (morphologically or biochemically or through molecular methods). Add reference for it as well.
· In whole article the words specie and strain is used interchangeably e.g Enterobacteriaceae strains and Enterobacteriaceae species. As these are two different things, it needs rectification.
· In the result section of the article, data is written as significant upon comparison but it (significance) has not been represented anywhere in figure.
· The discussion must be improved as only comparison is made at most instance rather discussing the philosophy behind any variation that occurred. In addition discussion in context of variation in occurrence of different species in samples needs to be added in discussion section.
· Overall the conclusion needs improvement as various findings are discussed in conclusion rather than deducing a suitable conclusion
· What is the impact of this study how would it help in improvement of public health.
Author Response
point-by-point responses to the Reviewer’s comments
Thank you for taking the time to review our manuscript and provide insightful and helpful comments, despite your busy schedule. We sincerely appreciate it. We have made revisions to the content of the paper based on your valuable comments. We kindly request you to review it once again. Thank you in advance for your assistance.
Reviewer2
Reviewer's suggestion or questions
1.The work is epidemiological in nature and the authors have basically emphasized on prevalence of different CRE species isolated from different types of submitted sample such as urine blood etc. only temporal distribution and proportion of different isolates are taken into account. An epidemiological survey include different aspects which must be taken into account such as demographics and geographic distribution. Wherein no such data is presented. Therefore for improving the article it is necessary to add such data.
Response to the reviewer
We agree that data on demographic and geographic distribution are important for epidemiological studies. Naturally, it is inferred that our data is likely to be influenced by densely populated areas. However, this study is based on aggregated data obtained from testing companies. Detailed individual information is not available for this dataset, and the regional characteristics were not analyzed.
Reviewer's suggestion or questions
- In abstract, line 10-11 needs rephrasing for better understanding.
Response to the reviewer
We have accordingly revised the sentence to enhance its clarity.
Reviewer's suggestion or questions
- Introduction provides a solid foundation for the research. I would like to suggest inclusion of additional content in the introduction to provide a more comprehensive context for the study.
Response to the reviewer
Thank you for the suggestion. I have described the current global situation for comparison and highlighted the differences from the situation in Japan.
Reviewer's suggestion or questions
- In introduction section, add transition sentences between paragraphs to improve the flow and connection of ideas.
Reviewer's suggestion or questions
We have accordingly revised the Introduction section to improve the flow.
Reviewer's suggestion or questions
- In Introduction, highlight the research gap that your study intends to address. What specific aspects of CRE epidemiology in smaller healthcare settings have not been explored in previous research?
Response to the reviewer
In Japan, it is often said that K. aerogenes is the most common causative agent of CRE infections. However, K. aerogenes is considered non-CP (carbapenemase-producing), so the risk of horizontal transmission is relatively low. However, in smaller healthcare settings, it has been observed in that carbapenem-resistant E. coli and K. pneumoniae are more prevalent as colonizers. Carbapenem-resistant E. coli and K. pneumoniae are known to possess carbapenemases, which can lead to horizontal transmission. This study has revealed the distribution of CRE on a national scale, including colonization, which had not been previously elucidated.
Reviewer's suggestion or questions
- In material method section, the author has not mentioned how the species are identified (morphologically or biochemically or through molecular methods). Add reference for it as well.
Response to the reviewer
The identification of bacteria was performed using an automated system, as described in the "Bacterial Database" section, using the Microscan WalkAway system. We have added details regarding the panel types and the method of inoculating bacterial cultures to ensure clarity. I have also included the necessary references.
Reviewer's suggestion or questions
- In whole article the words specie and strain is used interchangeably e.g Enterobacteriaceae strains and Enterobacteriaceae species. As these are two different things, it needs rectification.
Response to the reviewer
As per your suggestion, we have unified the term "isolate" instead of "strain."
Reviewer's suggestion or questions
- In the result section of the article, data is written as significant upon comparison but it (significance) has not been represented anywhere in figure.
Response to the reviewer
There was also feedback from other reviewers about the abundance of numerical data, so I have converted the table into a figure. Figure 1 does not include statistical significance. Regarding Figure 2, The statistical significance of the CRE percentages for E. cloacae and K. aerogenes is described in the legend of Figure. For Figure 3, as it was simplified to include only the major bacterial species, there is no need to include statistical significance.
Reviewer's suggestion or questions
The discussion must be improved as only comparison is made at most instance rather discussing the philosophy behind any variation that occurred. In addition discussion in context of variation in occurrence of different species in samples needs to be added in discussion section.
Response to the reviewer
We have accordingly revised the discussion section to discuss the observed trends in the occurrence of each bacterial species. We also clarify that our study primarily focuses on identifying the prevalent CRE species among specimens submitted to the testing company.
Reviewer's suggestion or questions
- Overall the conclusion needs improvement as various findings are discussed in conclusion rather than deducing a suitable conclusion
We have accordingly revised the conclusion section. This study relies solely on antimicrobial susceptibility testing to identify CRE, focusing on determining which bacterial species are more prevalent among the specimens submitted to the testing company. In the broader context, including colonization, E. coli and K. pneumoniae emerge as the central species associated with CRE, whereas E. cloacae and K. aerogenes, which are commonly reported as infectious agents, are comparatively low in terms of frequency.
Reviewer's suggestion or questions
- What is the impact of this study how would it help in improvement of public health.
Response to the reviewer
Our study reports for the first time that E. coli is the central species associated with CRE in Japan, based on large-scale investigations that are not publicly reported. Given its lower frequency in infectious cases, it is likely that E. coli has established itself more as a colonizer. Moreover, carbapenem resistance in E. coli is often attributed to carbapenemases, which raises concerns about horizontal gene transfer to other bacterial species. Considering the frequent movement of patients between healthcare facilities, monitoring colonization will also be necessary in the future.
Round 2
Reviewer 1 Report
The authors have done the suggested correction. The paper is acceptable in the present form.
Reviewer 2 Report
Article is sufficiently improved
Ok